# Leveraging the Cardio-Protective and Anticancer Properties of Resveratrol in Cardio-Oncology

**DOI:** 10.3390/nu11030627

**Published:** 2019-03-14

**Authors:** Ibrahim Y. Abdelgawad, Marianne K.O. Grant, Beshay N. Zordoky

**Affiliations:** Department of Experimental and Clinical Pharmacology, College of Pharmacy, University of Minnesota, Minneapolis, MN 55455, USA; abdel217@umn.edu (I.Y.A.); grant032@umn.edu (M.K.O.G.)

**Keywords:** resveratrol, cardio-oncology, cancer, cardiovascular, anthracycline, cardiotoxicity

## Abstract

Cardio-oncology is a clinical/scientific discipline which aims to prevent and/or treat cardiovascular diseases in cancer patients. Although a large number of cancer treatments are known to cause cardiovascular toxicity, they are still widely used because they are highly effective. Unfortunately, therapeutic interventions to prevent and/or treat cancer treatment-induced cardiovascular toxicity have not been established yet. A major challenge for such interventions is to protect the cardiovascular system without compromising the therapeutic benefit of anticancer medications. Intriguingly, the polyphenolic natural compound resveratrol and its analogs have been shown in preclinical studies to protect against cancer treatment-induced cardiovascular toxicity. They have also been shown to possess significant anticancer properties on their own, and to enhance the anticancer effect of other cancer treatments. Thus, they hold significant promise to protect the cardiovascular system and fight the cancer at the same time. In this review, we will discuss the current knowledge regarding the cardio-protective and the anticancer properties of resveratrol and its analogs. Thereafter, we will discuss the challenges that face the clinical application of these agents. To conclude, we will highlight important gaps of knowledge and future research directions to accelerate the translation of these exciting preclinical findings to cancer patient care.

## 1. Introduction

Cardio-oncology has emerged as a novel clinical/scientific discipline with a goal to prevent and/or treat cardiovascular diseases in cancer patients, particularly those arising as adverse effects of cancer treatment [1]. While cardio-oncology has recently been acknowledged as a clinical subspecialty [2], the cardiovascular adverse effects of cancer treatment were recognized more than 40 years ago. For instance, the cardiotoxicity of anthracyclines was reported in cancer patients in the early 1970s [3,4]. Despite the known cardiotoxic effects of these anticancer agents, they are still commonly used to treat a wide variety of malignancies in pediatric and adult cancer patients, due to their effectiveness as anticancer agents. Although every effort has been exerted to design safer anticancer agents, it seems that the occurrence of cardiovascular adverse effects during cancer treatment is inevitable, since newer anticancer drugs have also demonstrated significant cardiovascular adverse effects. Trastuzumab has been shown to cause cardiac dysfunction in breast cancer patients which was exacerbated if used in combination with an anthracycline [5,6]. The new proteasome inhibitor carfilzomib has been reported to cause cardiovascular adverse events in more than 18% of patients [7]. Similarly, cardiovascular adverse effects have been reported in cancer patients receiving tyrosine kinase inhibitors e.g., sunitinib [8] and immune checkpoint inhibitors e.g., nivolumab and ipilimumab [9]. In addition to chemotherapy, radiation, particularly chest radiation, has been shown to be detrimental to the cardiovascular system [10]. Therefore, there is an urgent need to identify new therapeutic agents that can prevent and/or reverse cancer treatment-induced cardiovascular adverse effects without compromising their anticancer therapeutic benefits.

Resveratrol (3,5,4′-trihydroxy-trans-stilbene) is a polyphenolic compound that is naturally occurring in several plant based foods and beverages including red wine, grapes, berries, and peanuts [11]. In addition, one of the richest sources of resveratrol is *Polygonum cuspidatum*, a herb that plays an important role in Chinese traditional medicine [12]. Indeed, resveratrol is a phytoalexin produced by more than 70 species of plants in response to stressful situations such as infection [13]. Resveratrol has been a focus of scientific research over the past 25 years, primarily as an active ingredient in red wine that is responsible for the “French Paradox” [14]. The “French Paradox” describes the low incidence of cardiovascular diseases in the French population despite high consumption of saturated fats [15]. Nevertheless, the concentration of resveratrol in red wine and other plants is very low and research into resveratrol has mostly used high doses that can only be achieved by supplementary intake of resveratrol as a nutraceutical [16]. Additionally, orally administered resveratrol shows very low bioavailability despite almost complete absorption (reviewed in [17]). Resveratrol low bioavailability is due to its first-pass intestinal/hepatic metabolism to sulfate and glucuronide metabolites [18]. After an oral dose of 25 mg in healthy human volunteers, the plasma concentration of resveratrol was less than 10 µg/L (≈40 nM), while the total concentration of resveratrol and its metabolites was 400–500 µg/L (≈2 µM) [18,19]. Higher peak plasma concentrations of resveratrol (approximately 500–2000 µg/L ≈2–10 µM) were achieved with doses of 2000–5000 mg [20,21,22]. Accordingly, the findings of in vitro studies using resveratrol concentrations higher than 10 µM should be interpreted with caution.

Preclinical in vitro and in vivo studies have revealed that resveratrol possesses a plethora of beneficial effects in an array of diseases (comprehensively reviewed in [23]), including: cardiovascular diseases (reviewed in [24,25,26]), cancer (reviewed in [27,28]), obesity (reviewed in [29]), osteoporosis (reviewed in [30]), Alzheimer’s disease (reviewed in [31]), and diabetes (reviewed in [32]). Resveratrol has also gained considerable attention as an anti-aging molecule (reviewed in [33]). It may be surprising that a single molecule can have beneficial effects in such a wide variety of diseases. Indeed, this may be attributed to the multiple molecular targets of resveratrol (reviewed in [34]). Resveratrol has been shown to activate AMP-activated protein kinase (AMPK) [35], sirtuin 1 (SIRT1) [36,37], super oxide dismutase (SOD) [38], nuclear factor erythroid 2–related factor 2 (NRF2) [39], vascular endothelial growth factor (VEGF) [40], and endothelial nitric oxide synthase (eNOS) [37,41]. It has also been shown to inhibit cyclooxygenases [42], phosphodiesterases (PDEs) [43], nuclear factor kappa B (NF-κB) [44,45], aryl hydrocarbon receptor (AhR) [46], phosphoinositide 3-kinase (PI3K) [47], mammalian target of rapamycin (mTOR) [48], and ribosomal protein S6K [49].

Since resveratrol has been shown to possess both cardio-protective and anticancer properties (Figure 1), leveraging these properties in cardio-oncology seems very promising. Indeed, a considerable amount of preclinical literature demonstrates that resveratrol protects against the cardiovascular adverse effects of a number of cancer treatments. Similarly, a plethora of in vitro and in vivo studies demonstrate that resveratrol enhances the anticancer effects of these treatments. Therefore, the objective of this review is to summarize and critically evaluate the contemporary knowledge that describes the effects of resveratrol when combined with cancer treatments on the cardiovascular system as well as on the tumor/cancer cells, in a cardio-oncology perspective. We will first discuss the interaction of resveratrol with anthracyclines, since this group of chemotherapeutic agents is considered to be the most cardiotoxic. Thereafter, we will discuss the existent knowledge about the effect of resveratrol on cardiovascular toxicity and chemotherapeutic efficacy of other cancer treatments that are known to be cardiotoxic. Moreover, we will discuss the potential cardio-protective and anticancer effects of the most common resveratrol analogs. To conclude, we will highlight significant gaps in knowledge and discuss important future research directions in order to translate this preclinical knowledge to benefit cardio-oncology patients. 

## 2. Anthracyclines

### 2.1. Resveratrol Protects against Anthracycline-Induced Cardiotoxicity

Anthracyclines (e.g., doxorubicin (DOX) and daunorubicin) are a group of chemotherapeutic agents used to treat a wide variety of human malignancies. However, the clinical use of these highly effective agents is limited by a significant anthracycline-induced cardiotoxicity which can progress to end-stage heart failure [50,51]. DOX has both acute and chronic toxic effects on the cardiovascular system. The acute effects occur in approximately 11% of patients during or soon after DOX administration and include various arrhythmias, hypotension, and acute heart failure [52,53]. On the other hand, chronic DOX cardiotoxicity is dose-dependent and results in irreversible cardiomyopathic changes that affect 1.7% of patients treated with DOX [54]. The exact mechanism of anthracycline-induced cardiotoxicity and its progression to heart failure has not been fully elucidated yet; however, several mechanisms have been proposed. These mechanisms include increased oxidative stress, mitochondrial dysfunction, apoptotic cell death, altered molecular signaling, and perturbed myocardial energy metabolism [52,55,56,57]. Interestingly, resveratrol has been shown to prevent all these detrimental changes, leading to significant protection against DOX-induced cardiotoxicity in preclinical studies in vitro (Table 1) and in vivo (Table 2).

DOX tends to generate reactive oxygen species (ROS) during its metabolism [84], largely because it is converted to an unstable semiquinone intermediate that favors ROS generation [85]. In addition, mitochondrial DNA damage induced directly by DOX or indirectly by DOX-generated ROS leads to respiratory chain failure and generation of more ROS [86]. DOX-induced oxidative stress is evident through increased levels of ROS, lipid peroxidation, and reductions in endogenous antioxidants and sulfhydryl group levels [87,88,89]. Resveratrol has been shown to abrogate DOX-induced oxidative stress by upregulating several antioxidant enzymes such as superoxide dismutase and catalase in vitro [58,60,62,66] and in vivo [68,70,79,81]. DOX-induced cardiotoxicity has also been associated with cardiomyocyte apoptosis [53]. DOX-generated ROS promotes the release of calcium from the sarcoplasmic reticulum, leading to an increase in intracellular calcium levels [90]. Mitochondria can capture a large quantity of the released calcium, eventually leading to the release of cytochrome c and apoptosis inducing factor [91,92]. In addition, DNA lesions induced directly by DOX or indirectly by DOX-generated ROS result in increased expression and activation of p53, which up-regulates genes such as the pro-apoptotic BAX [93]. Resveratrol has been shown to mitigate DOX-induced apoptosis to confer cardio-protection both in vitro [62,64] and in vivo [73,76,77]. 

Necrotic cell death has also been implicated in DOX-induced cardiotoxicity. Several studies have demonstrated that cardiac expression of pro-inflammatory cytokines, inflammatory cell infiltration, and necrotic cells are increased in the hearts of DOX-treated animals [53,94]. Mechanistically, DOX-generated ROS can lead to mitochondrial calcium overloading and opening of mitochondrial permeability transition pores which result in mitochondrial swelling, ATP depletion, and eventually necrotic cell death [95]. A few studies have shown that resveratrol protects against DOX-induced necrotic cell death [78,80]. Although a number of studies have shown that resveratrol protects against DOX-induced myocardial fibrosis [80,81,82], little is known about the effect of resveratrol on modulating the inflammatory response to DOX. Autophagy has also been shown to contribute to DOX-induced cell death [96]. However, the studies describing the effect of resveratrol on DOX-induced autophagy are inconsistent. On one hand, one study has shown that resveratrol prevents DOX-induced autophagic cell death [61]. On the other hand, a number of studies demonstrate that resveratrol induces autophagy to protect against DOX-induced apoptosis and cardiotoxicity [64,67]. This discrepancy may have arisen from the fact that autophagy can lead to either cell survival or cell death depending on the context [97]. Finally, cardiovascular senescence may also play a role in DOX cardiotoxicity as it has been shown that cultured neonatal rat cardiomyocytes treated with DOX exhibited characteristic changes similar to cardiomyocytes of aged rats [98]. Although resveratrol has been proposed as an anti-aging molecule [99], little is known about its effect on DOX-induced cardiovascular senescence. We have recently shown that the combined treatment of DOX followed by angiotensin II upregulated several senescence-associated genes including: p21, insulin-like growth factor binding protein 3 (Igfbp3), and growth arrest and DNA damage inducible gamma (Gadd45g) which was corrected by the coadministration of resveratrol with DOX [83].

Numerous studies have shown that DOX reduces cardiac energy reserves, particularly ATP and phosphocreatine, in different animal models of cardiotoxicity as well as in patients receiving DOX treatment [100,101,102]. This energetic deficit caused by DOX has been firstly attributed to compromised mitochondrial function [84]. More recently, DOX-induced cardiotoxicity has been shown to affect the phosphotransfer network of creatine kinase and the AMPK signaling pathway [103]. Intriguingly, resveratrol has been shown to ameliorate DOX-induced mitochondrial dysfunction [60,79] and to activate the AMPK signaling pathway [65], potentially leading to improved myocardial energetics and protection from DOX-induced cardiotoxicity.

### 2.2. Resveratrol Augments DOX-Induced Anticancer Effects

Protection against cancer treatment-induced toxicity is challenging because any protective treatment or strategy should not reduce the therapeutic benefit of the cancer treatment. In addition to its cardio-protective properties, resveratrol has also been shown to possess significant chemo-preventive and anticancer effects. Therefore, resveratrol holds the promise that it can protect the heart and fight the cancer at the same time. Since the anticancer properties of resveratrol have been discussed in a number of excellent reviews [27,104,105,106], we will focus here on the studies that demonstrated an additive and/or synergistic effect when resveratrol is combined with an anthracycline both in vitro (Table 3) and in vivo (Table 4).

A plethora of in vitro studies have shown that resveratrol enhanced the cytotoxic effect of DOX primarily in human breast cancer cells [107,109,110,111,112,114,116,119,121], but also in glioblastoma [107], prostate cancer [107], ovarian cancer [59,125], melanoma [108], hepatocellular carcinoma [109], cervical cancer [109], pancreatic cancer [122], leukemia [107,118], colon carcinoma [113,117], multiple myeloma, and lymphoma [118]. Since DOX is the most commonly used chemotherapeutic agent in canine hemangiosarcoma, we have recently shown that resveratrol can also augment the cytotoxic effect of DOX in two hemangiosarcoma cell lines [123]. Since resveratrol itself has been shown to induce apoptosis in cancer cells, it is not surprising that it had an additive proapoptotic effect when combined with DOX [108,115,118,123]. More intriguingly, however, is the finding that resveratrol increased cellular accumulation of DOX by inhibiting P-glycoprotein (P-gp), multidrug resistance-1 (MDR1), and multidrug resistance-associated protein-1 (MRP1) [109,117,122]. This mechanism has been most noticeable in DOX-resistant subclones such as MCF-7/ADR [112,114,125]. 

Since a major limitation for the clinical utility of resveratrol is its poor bioavailability, it is important to show that resveratrol can also exert anticancer effects when administered orally in vivo. Indeed, a number of studies have demonstrated that oral resveratrol administration suppressed tumor growth when given alone [128,129,130] and augmented the chemotherapeutic effect of DOX when given in combination to tumor-bearing mice in vivo [112,120,121]. A major limitation to the vast majority of these studies is the use of immunocompromised nude mice implanted with human cancer cells (Table 4). This approach precludes studying the immunomodulatory properties of resveratrol which are increasingly recognized to be important players in its chemo-preventive and anticancer effects [131,132,133].

## 3. Other Cardiotoxic Cancer Treatments

### 3.1. Cisplatin

Cisplatin is a first-generation platinum-based chemotherapeutic agent. Unlike anthracyclines, cisplatin-induced cardiotoxicity is uncommon, and its exact prevalence is unknown [134]. Preclinical studies demonstrated that cisplatin-induced cardiotoxicity may be mediated by mitochondrial abnormalities, increased endoplasmic reticulum stress, oxidative stress, and apoptosis [135,136]. In contrast to the large number of studies that reported cardio-protective effects of resveratrol against anthracycline-induced cardiotoxicity, a single study was found on PubMed that reported the effect of resveratrol on cisplatin-induced cardiotoxicity. In this study, oral administration of resveratrol (5, 15, or 45 mg/kg/day for 10 days) protected against cisplatin-induced cardiotoxicity in adult male Wistar rats by alleviating cisplatin-induced oxidative stress [136]. However, resveratrol has been shown to protect against cisplatin-induced nephrotoxicity [137,138,139], gonadotoxicity [140,141,142], and ototoxicity [143,144,145]. Pretreatment with resveratrol did not significantly alter the plasma level of cisplatin; however, it lowered its urinary concentration and its accumulation in the kidneys, which may explain its protective effects [146]. 

Importantly, resveratrol has also been shown to potentiate the cytotoxic effect of cisplatin in vitro via different mechanisms including: glutamine metabolism inhibition in human hepatoma cells [147], increased mitochondrial oxidative damage in malignant mesothelioma cells [148], enhanced autophagic cell death in A549 cells [149], inducing the mitochondrial apoptotic pathway in human non-small cell lung cancer H838 and H520 cell lines [150], increasing the cellular uptake of cisplatin in parent and cisplatin-resistant HCT-116 colorectal cancer cells [151], induction of dual specificity phosphatase 1 in androgen-independent prostate cancer cells [152], preventing DNA repair of double-strand breaks [153] and phosphorylation of p53 at the serine 20 position in human breast cancer cells [154], and modulation of intrinsic apoptosis in H446 small cell lung cancer [155]. Resveratrol significantly enhanced the antineoplastic effect of cisplatin in a mouse model of ovarian cancer, through inhibition of glucose uptake in ovarian tumor cells with high baseline glycolytic rates [156].

### 3.2. Cyclophosphamide

Cyclophosphamide is an alkylating agent that is clinically used in the management of autoimmune conditions and a number of malignant diseases like lymphoma, leukemia, and breast cancer [157]. Although conventional doses of cyclophosphamide are rarely associated with cardiotoxicity, symptomatic cardiotoxicity occurs in 5–28% of patients treated with high doses [158,159]. Cyclophosphamide-induced cardiotoxicity presents as myocarditis, pericarditis, cardiomyopathy, reversible myocardial failure, and/or isolated arrhythmias [159,160]. Cyclophosphamide-induced oxidative and nitrosative stress are believed to mediate its cardiotoxicity [161]; however, the exact mechanism is not fully elucidated [157]. Resveratrol (10 mg/kg/day for 8 days) has been reported to protect against acute cyclophosphamide-induced cardiotoxicity, primarily via mitigation of cyclophosphamide-induced oxidative stress [162]. Importantly, resveratrol (50 µM) enhanced the antiproliferative and apoptotic effect of cyclophosphamide in MCF-7 breast cancer cells [163,164].

### 3.3. Arsenic Trioxide

Arsenic trioxide is a chemotherapeutic agent that is mainly used in the treatment of relapsed or refractory acute promyelocytic leukemia [165]. Its clinical utility has been limited by multiorgan toxicity, particularly cardiotoxicity [166]. Arsenic trioxide-induced cardiotoxicity causes QT prolongation which may lead to ventricular tachycardia, torsade de pointes, and sudden death [167]. The molecular mechanism underlying this adverse effect is believed to be arsenic trioxide-induced increase in calcium currents to cardiomyocytes [168]. In addition, arsenic trioxide has been shown to induce oxidative stress, inflammation, and apoptosis (reviewed in [166]). In a BALB/c mouse model of arsenic trioxide-induced cardiomyopathy in vivo, resveratrol was administered intravenously 3 mg/kg every other day for 3 doses, 1 h before arsenic trioxide administration. In this model, resveratrol ameliorated arsenic trioxide-induced QT interval prolongation, oxidative damage, and cardiomyocyte injury (apoptosis, myofibrillar loss, and vacuolization) [169]. Similarly, in Wistar rats, resveratrol was administered intravenously 8 mg/kg every other day for 4 doses, 1 h before arsenic trioxide administration. Pretreatment with resveratrol ameliorated arsenic trioxide-induced oxidative stress, intracellular calcium accumulation, pathological alterations, and cardiac dysfunction [170]. These protective effects were associated with an induction of antioxidant enzymes and a decrease in myocardial arsenic concentration [170]. In vitro studies have also shown that resveratrol confers protection against arsenic trioxide damage. In H9c2 cardiomyoblasts, resveratrol (1–10 µM) protected against arsenic trioxide-induced oxidative stress, apoptosis, and accumulation of intracellular calcium [169]. In guinea pig ventricular cardiomyocytes, resveratrol ameliorated arsenic trioxide-induced damage of the human ether-a-go-go-related gene (hERG) current, relieved endoplasmic reticulum stress, and shortened the action potential duration [171]. Resveratrol also had a protective effect against arsenic trioxide-induced oxidative damage in the feline brain, lung, and kidney [172,173,174], and nephrotoxicity and hepatotoxicity in rats [175,176]. 

Resveratrol enhanced the cytotoxic effect of arsenic trioxide in several cancer cell lines, including: human lung adenocarcinoma A549 cells, SK-N-SH neuroblastoma cells, lung adenocarcinoma A549 cells, MCF-7 breast cancer cells, and promyelocytic leukemia NB4 cells through increased ROS and augmented apoptotic response [177,178,179,180,181]. In BALB/c nude mice, treatment with resveratrol (16.5 mg/kg/day), arsenic trioxide (5 mg/kg/day), or the combination for 2 weeks caused significant reduction in tumor growth; however, there was no noticeable difference between the combination therapy and either drug given alone [181]. Intriguingly, Fan et al. concurrently studied the chemosensitizing and cardio-protective effects of resveratrol when combined with arsenic trioxide in vitro. Resveratrol (5 µM) increased ROS and potentiated arsenic trioxide cytotoxicity in human promyelocytic leukemia NB4 cells, whereas it abrogated arsenic trioxide-induced ROS and protected against its toxic effect in neonatal rat ventricular cardiomyocytes [178]. 

### 3.4. Tyrosine Kinase Inhibitors

Tyrosine kinase inhibitors (e.g., sunitinib, dasatinib, imatinib, and nilotinib) are a relatively newer group of chemotherapeutic agents that predominantly target tyrosine kinase enzymes. They are used in treatment of several cancers including leukemia, renal cell carcinoma, hepatocellular carcinoma, and melanoma [157]. The occurrence of cardiovascular adverse effects varies among these agents, with sunitinib showing the worst cardiovascular safety profile [182]. Sunitinib is associated with significantly increased risk of heart failure in up to 8%, and hypertension in up to 50% of patients [183]. The main mechanism of sunitinib-induced cardiotoxicity is the inhibition of off-target kinases, such as mitogen-activated protein kinase kinase 7 [184], mitogen-activated protein kinases (p38, JNK, and ERK1/2) [185], and AMP-activated protein kinase (AMPK) [186]. Activation of the AhR signaling pathway has also been reported to mediate sunitinib-induced cardiac hypertrophy in vivo and in vitro [187]. Interestingly, resveratrol (20 µM) prevented sunitinib-induced H9c2 cardiomyoblast hypertrophy and abrogated sunitinib-mediated induction of CYP1A1, an AhR-regulated gene [187]. Conversely, resveratrol did not prevent dasatinib-mediated induction of hypertrophic markers or CYP1A1 genes in H9c2 cardiomyoblasts [188]. 

There is still a paucity of research determining the impact of resveratrol on the cytotoxic effect of tyrosine kinase inhibitors. Although resveratrol has been shown to prevent sunitinib-mediated induction of CYP1A1 in MCF-7 breast cancer cells, the effect of resveratrol on sunitinib-induced cytotoxicity in these cancer cells was not reported [189]. In chronic myeloid leukemia cells, resveratrol increased dasatinib-, nilotinib-, and imatinib-induced apoptosis [190,191,192]. Interestingly, resveratrol alone showed significant cytotoxic effect in imatinib-resistant chronic myeloid leukemia cells [192,193].

### 3.5. Radiation

Radiation may induce a number of cardiovascular adverse effects, including: pericarditis, pericardial fibrosis, myocardial fibrosis, coronary artery disease, and microvascular damage (reviewed in [194]). The mechanisms of radiation-induced cardiovascular toxicity include: increased oxidative stress [195,196], inflammation [197,198], apoptotic cell death [199], and cellular senescence [200]. Although resveratrol has the potential to prevent all these detrimental effects, it is rarely studied as a protective agent against radiation-induced cardiovascular toxicity. In a single study, black grape juice with high resveratrol content has been shown to abrogate radiation-induced oxidative stress in the heart tissue of rats [201]. Although this study showed that black grape juice normalized the increase of lactate dehydrogenase, which can be a marker of cardiac damage, it did not report the protective effect on cardiac function or structure [201]. Beyond cardiovascular protection, resveratrol has been shown to protect against radiation-induced erectile dysfunction in male rats [202], immunosuppression in mice [203], intestinal injury in mice [204], hepatotoxicity in rats [205], ovarian toxicity in female rats [206,207], salivary gland damage in mice [208] and rats [209], and hematopoietic stem cell injury in mice [210].

The potential use of resveratrol to protect against radiation-induced cardiovascular toxicity is intriguing, because resveratrol has been largely shown to sensitize cancer cells to radiation (Table 5). The radiomodulatory effect of resveratrol was determined in CHO-k1 and A549 cell lines, wherein the low concentration of resveratrol, 15 µM, protected from radiation-induced genotoxic damage, while the high concentration, 60 µM, augmented radiation-induced genotoxic damage. It is important to mention that while the low concentration protected against genotoxic damage, it did not reduce the cytotoxic effect of radiation as assessed by the MTT cell viability assay [211]. Resveratrol (10–100 µM) and irradiation with 4 Gy alone and in combination significantly decreased cell viability in rodent GH3 and TtT/GF pituitary adenoma cells in vitro [212]. Resveratrol (10 μM) sensitized the human breast cancer cell line MCF7 to the cytotoxic and anti-proliferative effect of 3 Gy ionizing radiation [213]. Supra-additive cytotoxic effect was observed when cervical squamous cell cancer cell line (HeLa) was treated with resveratrol at its CC50 followed by radiotherapy at 2Gy. The same effect was also observed with other mTOR inhibitors including temsirolimus, everolimus, curcumin, and epigallocatechin gallate [214]. Conversely, resveratrol (50 μM) protected lung cancer cell lines A549 and H460 against radiation-induced apoptosis through Sirt1 activation [215].

## 4. Resveratrol Analogs

There are other naturally occurring stilbene-like compounds related to resveratrol that have shown promising anticancer properties (reviewed in [237]). In addition, a number of resveratrol analogs have been synthesized to enhance the bioavailability and pharmacologic properties of resveratrol (reviewed in [238,239]). These analogs have also demonstrated promising chemo-preventive and anticancer properties in preclinical studies (reviewed in [237]). 

### 4.1. Pterostilbene 

Pterostilbene (3,5-di-methoxy-4′-hydroxystilbene) is a dimethylated analog of resveratrol that has been extensively studied as a potential chemo-preventive and therapeutic agent against different types of cancer [106,240,241,242]. It has been shown to enhance the chemotherapeutic effect of some potentially cardiotoxic chemotherapies. For instance, pterostilbene demonstrated synergistic anti-proliferative activity with cisplatin in several ovarian cancer cell lines [243]. However, the cardiovascular effects of pterostilbene are not as extensively studied as resveratrol. A few recent studies have suggested that pterostilbene may confer cardio-protection against myocardial ischemia-reperfusion injury through AMPK and SIRT1 activation [244,245], reduction in oxidative stress [246], and anti-inflammatory and anti-apoptotic effects [246,247]. It has also been shown to prevent cardiac hypertrophy and restore right ventricular function in a rat model of cor pulmonale [248]. Nevertheless, there are no studies that report the potential protective effect of pterostilbene against cancer treatment-induced cardiotoxicity. 

### 4.2. Tetramethoxystilbenes 

DMU-212 (3,4,5,4′-tetramethoxy-trans-stilbene) is a resveratrol analog that has been shown to be more potent than resveratrol in inhibiting the growth of several cancer cell lines such as breast, ovarian, and melanoma cells [249,250,251,252]. PicMet (3,5,3′,4′-tetramethoxystilbene) is another tetramethoxystilbene that has been shown to enhance the cytotoxic effect of DOX and increased its intra-cellular accumulation in DOX-resistant human adenocarcinoma cell line (LoVo/Dx) [253] and the L5178 mouse lymphoma cell line expressing the human MDR1 gene [254]. TMS (2,4,3′,5′-tetramethoxystilbene) is a selective inhibitor of cytochrome P450 CYP1B1, an enzyme that is overexpressed in some tumors and has protumorigenic activity [255]. TMS has been shown to inhibit cell viability of human breast cancer, leukemia, and prostatic cancer cell lines, largely by inhibiting CYP1B1 [256,257]. Intriguingly, TMS has recently been shown to protect against DOX-induced cardiotoxicity by inhibiting cytochrome P450 CYP1B1 and midchain hydroxyeicosatetraenoic acid (HETE) formation [258]. It has also been shown to protect against isoproterenol- and angiotensin II-induced cardiac hypertrophy [259,260], angiotensin II-induced aortic aneurysm [261], and atherosclerosis and hypertension [262,263] through inhibition of CYP1B1 enzyme. 

### 4.3. Trimethoxystilbene 

Trimethoxystilbene (trans-3,5,4′-trimethoxystilbene) is another resveratrol analog that also has promising anticancer properties, reviewed in [264]. However, its potential cardio-protective properties have rarely been studied. In a single study, it has been shown to prevent pulmonary vascular remodeling and right ventricle hypertrophy in a hypoxia-induced rat model of pulmonary arterial hypertension, which contributed to inhibition of oxidative stress and inflammation [265].

### 4.4. Piceatannol 

Piceatannol (trans-3,4,3′,5′-tetrahydroxystilbene) is a naturally occurring stilbene commonly found in grape skins and wine which has potential chemo-preventive and anticancer properties (reviewed in [266]). With regard to its potential cardiovascular benefits, piceatannol has been shown to prevent isoproterenol-induced cardiac hypertrophy [267]. Although piceatannol was shown to possess anti-arrhythmic properties [268], recent reports have shown that it does not have such properties [269] and it may even be pro-arrhythmic [270]. There are no reports of cardio-protective effects of piceatannol against cancer treatment-induced cardiotoxicity. Nevertheless, there are a number of studies showing that it can potentiate the chemotherapeutic effect of cardiotoxic chemotherapies. For instance, it potentiated the apoptotic effect of DOX in DOX-resistant human adenocarcinoma cell line (LoVo/Dx), primarily by inhibiting MDR protein and increasing the intracellular accumulation of DOX [253]. It also enhanced cisplatin sensitivity in ovarian cancer cells by acting on p53, x-linked inhibitor of apoptosis (XIAP), and mitochondrial fission [271]. 

### 4.5. Epsilon-Viniferin

Epsilon-viniferin (ε-viniferin) is a resveratrol dimer that has been shown to be more effective than resveratrol in reducing systolic blood pressure and mitigating cardiac hypertrophy in spontaneously hypertensive rats [272]. It enhanced cisplatin-induced apoptosis in C6 cells [273].

## 5. Challenges to the Clinical Use of Resveratrol in Cardio-Oncology

Despite all the aforementioned studies that support the use of resveratrol in cardio-oncology, there are several challenges to its clinical application in cancer patients receiving cardiotoxic cancer treatments. First, all the previous studies showing cardio-protective effects of resveratrol are preclinical studies conducted either in vitro or in rodent models of human diseases. Unfortunately, a number of agents which had shown significant cardio-protective properties in animals failed in subsequent clinical trials (e.g., vitamin E and N-acetyl cysteine [274,275]). These failures hampered the clinical interest in translating new cardio-protective agents into clinical trials. In clinical trials of resveratrol in cancer patients, modest beneficial effects were demonstrated at the molecular level in colon cancer [276] and breast cancer patients [277]. However, there were very disappointing results in patients with multiple myeloma [278]. Indeed, there was an unexpected renal toxicity in multiple myeloma patients who received SRT501 (micronized resveratrol, 5 g/day). This safety concern was enough to terminate the clinical trial [278]. Moreover, cancer patients are already receiving multiple chemotherapeutic agents ± radiation therapy. Therefore, the oncologist is usually very reluctant to add another drug that may interfere with the therapeutic benefit and/or increase the toxicity of these therapies. For instance, the fear of reducing the therapeutic benefit has hindered the use of dexrazoxane, the only FDA approved drug to prevent anthracycline cardiotoxicity. In one study, dexrazoxane has been shown to be associated with higher rates of secondary malignancy in childhood cancer survivors [279]. Although the results of this study were not corroborated by other studies [280,281], it was enough to hinder more widespread use of dexrazoxane. Furthermore, the predictive and diagnostic value of different approaches such as strain echocardiography, cardiac magnetic resonance imaging, and biomarkers are still not well established in the field of cardio-oncology [282]. Therefore, it is difficult to identify those patients who will benefit the most from a potential cardio-protective therapy.

In addition to these general concerns, resveratrol has its own particular challenges. First, its exact mechanism of action is not known. It has multiple targets that make it extremely difficult to predict how it will interact with the human body, different disease states, and different medications. Another major limitation of resveratrol is its poor bioavailability due to rapid metabolism to its sulfated and glucuronide metabolites [18]. Administration of up to 5 g of resveratrol yielded a plasma concentration of only 500 ng/mL (2 μM) in humans [22]. Despite this poor bioavailability, oral administration of resveratrol to experimental animals was almost as effective as its parenteral administration, creating the “resveratrol paradox” which describes resveratrol as a molecule with very low plasma level, but significant biological effects [283]. A number of mechanisms have been postulated to explain this phenomenon including higher tissue concentration of resveratrol in certain organs [284], conversion of these metabolites back to resveratrol [285], and biological activity exerted by these metabolites [286]. More recently, it has been shown that resveratrol may exert some of its beneficial effects through alteration of the gut microbiome [287,288,289], a mechanism that does not need systemic absorption. It is also important to mention that dietary intake of resveratrol is not likely to result in the aforementioned pharmacological effects, since the average daily intake of resveratrol is estimated to be less than 1 mg [290]. 

## 6. Conclusions

There is an obvious mismatch between the plethora of preclinical studies and the limited number of clinical studies of resveratrol that support its use in the field of cardio-oncology. Despite the very promising preclinical findings of resveratrol as a cardio-protective agent, there are still several questions that need to be answered before advancing resveratrol into clinical trials. First and foremost, all the previously reviewed studies showed the cardio-protective effect of resveratrol in tumor-free experimental animals. Although resveratrol has been shown to augment the chemotherapeutic effect of these cardiotoxic cancer treatments, no study has yet demonstrated the cardio-protective and the anticancer effects of resveratrol simultaneously in tumor-bearing models. Taking into account the possible hormetic dose-response properties of resveratrol [291], it is extremely important to show that the same dose of resveratrol is able to protect the heart and fight cancer. Second, it is also important to determine the effect of resveratrol on the pharmacokinetics and tissue distribution of chemotherapeutic agents, taking into account the potential of resveratrol to alter a number of drug metabolizing enzymes [292]. Indeed, canine cancer patients can offer a perfect translational platform to answer these questions. Resveratrol is a common nutritional supplement in dogs and it has been shown to have cytotoxic effects in canine cancer cells [123,293]. Therefore, initiation of clinical trials of resveratrol in canine cancer patients seems a logical first step in translating the preclinical findings to human cancer patients. Finally, the emergence of novel resveratrol analogs with improved bioavailability and enhanced pharmacological properties could open new venues for preclinical testing of these analogs as new armaments in the battle of cardio-oncology.

## Figures and Tables

**Figure 1 nutrients-11-00627-f001:**
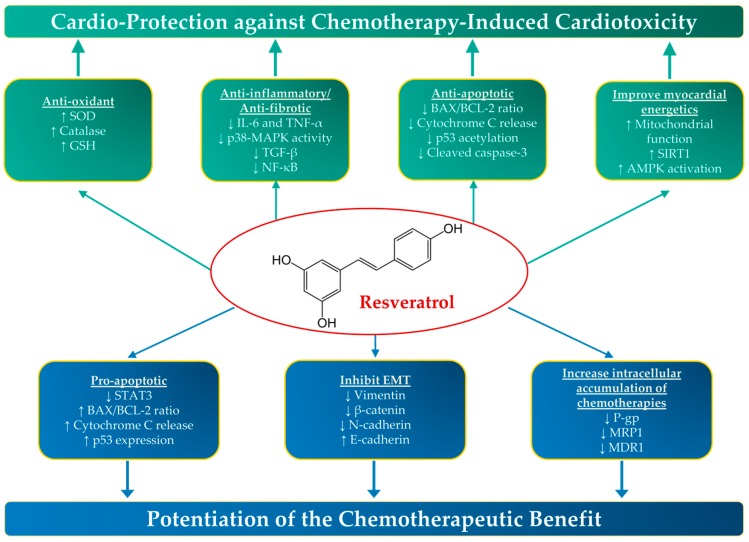
The cardio-protective and anticancer properties of resveratrol. Resveratrol has been shown to possess both cardio-protective and anticancer properties in preclinical studies through a number of molecular mechanisms. AMPK, Adenosine Monophosphate-Activated Protein Kinase; Bax, BCL (B Cell Lymphoma)-Associated X; EMT, Epithelial Mesenchymal Transition; GSH, Glutathione; IL-6, Interleukin-6; MAPK, Mitogen-activated protein kinase; MDR1, Multidrug Resistance-1; MRP1, Multidrug resistance-associated protein-1; NF-κB, nuclear factor kappa B; P-gp, P-glycoprotein; STAT3, Signal transducer and activator of transcription 3; SIRT1, Sirtuin 1; SOD, Superoxide Dismutase; TNF-α, Tumor Necrosis Factor-alpha; ↑, increase; and ↓, decrease.

**Table 1 nutrients-11-00627-t001:** In vitro studies demonstrating the cardio-protective effects of resveratrol against anthracycline-induced cardiotoxicity.

Study	Cell Type	DOX Treatment (Conc./Time)	Resveratrol Treatment (Conc./Time/Time Relative to DOX)	Finding	Proposed Mechanism
Cao Z et al. 2004 [58]	Rat cardiomyoblast H9c2 cells	5–40 μM for 24 h	100 μM for 72 h pre-DOX	RESV afforded marked protection against DOX-mediated cytotoxicity in H9c2 cells	↑ Antioxidants (SOD) and Phase 2 enzymes
Rezk YA et al. 2006 [59]	Neonatal rat ventricular cardiomyocytes	10 nM–100 μM	1 μM 1 h pre-DOX	RESV reduced DOX-induced cell death	Antioxidant properties
Danz ED et al. 2009 [60]	Neonatal rat ventricular cardiomyocytes	1 μM for 24 h	10 μM for 72 h	RESV protected against DOX-induced oxidative stress and subsequent cell death	↑ Mitochondrial function ↑ Sirt1 activation ↓ Mitochondrial ROS production ↑ SOD activity
Xu X et al. 2012 [61]	Neonatal rat ventricular cardiomyocytes	1 μM for 18 h	10 μM for 12 h pre-DOX	RESV protected against DOX-induced toxicity	↓ DOX-induced Autophagy ↓ S6K1 activity ↓ AKT signaling ↓ mTOR signaling
De Angelis A et al. 2015 [62]	Human Cardiac Progenitor cells (hCPCs)	0.5 μM for 24 h	0.5 μM co-treatment for 24 h	RESV protected hCPCs from DOX-induced death	↑ SIRT1 expression ↓ ROS generation ↑ SOD expression ↑ Catalase activity ↓ Apoptosis ↓ p53 acetylation
Lou Y et al. 2015 [63]	Rat cardiomyoblast H9c2 cells	5 μM for 24 h	0, 10, 25, 50 or 75 μM for 24 h followed by DOX	RESV protected H9c2 cells against DOX-induced apoptosis	↓ ER stress marker expression ↑ Sirt1 levels
Gu J et al. 2016 [64]	Rat cardiomyoblast H9c2 cells	2 μM for 24 h	20 μM for 24 h	Co-treatment strategy with RESV attenuated the cardiotoxic effects of DOX	↓ Apoptosis with slight ↑ in autophagy p-AMPK activation ↑ LC3 (particularly LC3-II) ↑ BCL-2 ↓ BAX expression
Liu MH et al. 2016 [65]	Rat cardiomyoblast H9c2 cells	5 µM for 24 h	25 μM for 24 h (Started 30 min pre-DOX)	RESV protected H9c2 cells from DOX-induced apoptosis	↓ BAX ↑ Cell viability ↑BCL-2 ↓ p53 induced-expression ↑ phosphorylation of AMPK
Liu MH et al. 2016 [66]	Rat cardiomyoblast H9c2 cells	5 µM for 24 h	25 μM for 24 h (Started 24 h pre-DOX)	RESV protected H9c2 cells against DOX-induced apoptosis	↑ Sirt1 activation ↑ SOD ↓ MDA expression
Gu J et al. 2018 [67]	Starved Rat cardiomyoblast H9c2 cells	1 μM for 24 h	20 μM for 24 h	RESV attenuated DOX-induced cytotoxicity	Blocking induction of E2F1/mTORC1 and E2F1/AMPKα2 pathway by DOX, leading to ↑ autophagy and ↓ apoptosis

AKT, Protein Kinase B; AMPK, AMP-activated protein kinase; BAX, BCL (B Cell Lymphoma)-Associated X; DOX, Doxorubicin; E2F1, E2F Transcription factor 1; ER, Endoplasmic reticulum; hCPCs, Human Cardiac Progenitor cells; LC3, Light chain 3; MDA, Malonaldehyde; mTORC1, Mammalian target of Rapamycin Complex 1; RESV, Resveratrol; ROS, Reactive oxygen species; S6k1, Ribosomal protein S8 kinase beta-1; Sirt, Sirtuin; SOD, Superoxide Dismutase; ↑, increase; and ↓, decrease.

**Table 2 nutrients-11-00627-t002:** In vivo studies demonstrating the cardio-protective effect of resveratrol against anthracycline-induced cardiotoxicity.

Study	Species/Strain	DOX Treatment (Dose/Time)	Resveratrol Treatment (Dose/Time/Time Relative to DOX)	Finding	Proposed Mechanism
**Models of Acute Cardiotoxicity**					
Wang GY et al. 2007 [68]	Adult male SD rats	A single dose of 10 mg/kg	30, 120 mg/kg i.p. once daily for 3 days	RESV relieved DOX-induced toxic effects on the myocardium	↓ Oxidative stress ↓ MDA and NO
Olukman M et al. 2009 [69]	Male Wistar rats	A single i.p dose of 20 mg/kg	10 mg/kg/i.p. concomitant with DOX	RESV reversed DOX-induced vascular endothelial dysfunction	Prevention of excessive NO formation
Osman AM et al. 2013 [70]	Male Wistar rats	A single dose of 20 mg/kg	10 mg/kg i.p. concomitant with DOX	RESV protected against DOX-induced cardiotoxicity	Antioxidant properties ↑ SOD activity
Pinarli FA et al. 2013 [71]	Female Wistar albino rats	A single i.p. dose of 20 mg/kg	100 mg/kg i.p. three times weekly (first 1 week before then at weekly intervals after DOX injection)	RESV protected against DOX-induced hemodynamic and histological changes	Antioxidant and anti-inflammatory properties
Al-Harthi SE et al. 2014 [72]	Male Wistar albino rats	A single dose of 20 mg/kg for 48 h	10 mg/kg i.p. concomitant with DOX	RESV ameliorated the effect of DOX on cardiac tissue	↓ MDA production
Ruan Y et al. 2015 [73]	Male C57BL/6J mice	A single i.p. dose of 20 mg/kg	10 mg/kg/day started 3 days pre-DOX and 5 days after	RESV prevented DOX-induced cardiac dysfunction	SIRT1 Activation ↓ Apoptosis and oxidative stress Inhibit caspase-3, Bax, and p38 MAPK activation ↑ BCL-2 and SOD-1
Sin TK et al. 2015 [74]	Young (2 months) and old (10 months) senescence-accelerated male mice prone 8 (SAMP8)	A single i.p. dose of 18 mg/kg	20 mg/kg/day for 3 days (Started one day after DOX administration)	RESV abrogated DOX-induced impairment of cardiac systolic function	Restoration of SIRT1 activity ↑ Level of p300 and ubiquitinated proteins in aged hearts. In young mice, ↓ catabolic signaling measured as acetylated Foxo1 and MuRF-1 induced by DOX
Gu J et al. 2016 [64]	Sprague–Dawley rats	A single i.p. dose of 15 mg/kg/day	10 mg/kg/day i.p.	RESV treatment attenuated apoptotic cell death and decreased the cardiotoxicity of DOX	↑ LC3 (particularly LC3-II) ↓ Cleaved caspase-3
**Models of Chronic Cardiotoxicity**					
Rezk YA et al. 2006 [59]	Severe combined immunodeficient mice	2 mg/kg i.v. every other day for 5 days	3 mg/kg, i.p. started 2 h pre-DOX and 1 h after	RESV pretreatment significantly ameliorated DOX-induced bradycardia and QTc prolongation	Antioxidant properties
Tatlidede E et al. 2009 [75]	Both sexes Wistar albino rats	Cumulative dose 20 mg/kg, i.p. 6 times for 2 weeks	10 mg/kg/day, i.p. for 7 weeks	RESV alleviated DOX-induced oxidative damage of the heart	↑ Antioxidant capacity ↓ Lipid peroxidation ↓ Neutrophil infiltration ↓ MDA level ↑ GSH level ↑ SOD and catalase activities
Zhang C et al. 2011 [76]	Male BALB/c Mice	8 mg/kg i.p. at 3-week intervals (cumulative dose of 24 mg/kg)	15 mg/kg/day (added to mouse standard chow for 7 weeks)	RESV protected against DOX-induced cardiomyocyte apoptosis	↑ SIRT1 expression ↓ p53 acetylation ↓ MDA levels ↓ Apoptosis ↓ Bax overexpression ↓ Cytochrome c release from mitochondria
Gu J et al. 2012 [77]	Male BALB/c nude Mice with induced lymphoma	Cumulative dose of 12 mg/kg, i.p. for 6 times for 2 weeks	10 mg/kg/day, p.o. for 3 weeks; started one-week pre-DOX	RESV protected the hearts against DOX-induced apoptosis and damage	↑ HO-1 expression and activity ↓ p53 expression ↓ Bax expression ↑ Bcl-2 expression ↓ Caspase-3 activity
Dudka J et al. 2012 [78]	Male albino rats of Wistar CRL: (WI)WUBR	1 or 2 mg/kg i.p. once weekly for 7 weeks	20 mg/kg of feed (concomitant administration, started one-week pre-DOX)	RESV reduced the increased lipid peroxidation caused by the 1 mg DOX dose, but it slightly intensified adverse cardiac histological changes. RESV attenuated necrosis and other cardiac histopathological changes induced by the 2 mg DOX dose	↓ Necrosis ↓ Inflammatory infiltration
Dolinsky VW et al. 2013 [79]	Female C57BL/6 mice	8 mg/kg i.p. once weekly for 4 weeks	4 g/kg diet, ad libitum concomitantly with DOX for 4 weeks	RESV prevented DOX-induced LV remodeling and reduced DOX-induced oxidative stress	↓ Lipid peroxidation ↑ Antioxidant defenses ↑ Mitofusin-1 and -2 expressions
Arafa MH et al. 2014 [80]	Adult male Wistar albino rats	2.5 mg/kg i.p. in six injections over 2 weeks (cumulative Dose of 15 mg/kg)	20 mg/kg Daily for 4 weeks (Started 2 weeks pre-DOX)	RESV protected against DOX-induced cardiotoxicity and fibrosis	↓ LV lipid peroxidation ↑ LV GSH and SOD ↓ TGF-β1 gene expression ↓ Necrosis and fibrosis
De Angelis A et al. 2015 [62]	Adult female Fischer 344 rats	2.5 mg/kg i.p. for 6 times over 2 weeks (cumulative dose of 15 mg/kg)	2.5 mg/kg concomitantly with DOX for 2 weeks + additional 4 weeks	RESV increased survival and improved cardiac function in animals with DOX-induced heart failure	↓ CPCs apoptosis
Cappetta D et al. 2016 [81]	Adult female F344 rats	2.5 mg/kg i.p. for 6 times over 2 weeks (cumulative dose of 15 mg/kg)	2.5 mg/kg/day by oral gavage (concomitant with DOX then additional 1 week)	RESV ameliorated end-diastolic pressure/volume relationship ↓ fibrosis and fibroblast activation	SIRT1 activation ↓ TGF-β levels ↓ p53 levels ↓ pSMAD3/SMAD3 levels ↑ Mn-SOD
Shoukry HS et al. 2017 [82]	Adult male Wister rats	2.5 mg/kg i.p. for 6 times over 2 weeks (cumulative dose of 15 mg/kg)	Prophylactic: 20 mg/kg for 6 weeks started 2 weeks with DOX then 4 weeks. Therapeutic: 20 mg/kg for 4 weeks started after DOX	Both prophylactic and therapeutic use of RESV mitigated DOX-induced deterioration of cardiac function with the prophylactic approach being more effective than therapeutic	↓ Myocardial apoptosis and fibrosis ↓ Bax expression ↓ NFAT3 expression
Matsumura N et al. 2018 [83]	Juvenile male C57BL/6 mice	4 mg/kg i.p. once weekly for 3 weeks	0.4% RESV in chow diet 1 week before, during, and 1 week after DOX administration	RESV normalized molecular markers of cardiotoxicity and restored the heart ability of adaptive remodeling in response to hypertension	Normalized DOX-induced cardiac p38 MAPK activation Corrected DOX-induced p53-regulated genes (p21, Igfbp3, Gadd45g, Perp) in response to AngII-induced hypertension
Gu J et al. 2018 [67]	Adult male C57BL/6 mice	Normal mice 4 times for 4 weeks (cumulative dose of 20 mg/kg)	10 mg/kg i.p. prior to each DOX injection	RESV attenuated DOX-induced cardiotoxicity	RESV inhibited the E2F1/AMPKα2 and E2F1/mTORC1 pathway induced by DOX in both models
		AMI group 5 mg/kg/day i.p. for 7 days	10 mg/kg/day i.p. prior to each DOX injection		

Akt, Protein Kinase B; AMPK, Adenosine Monophosphate-Activated Protein Kinase; AngII, Angiotensin II; Bax, BCL (B Cell Lymphoma)-Associated X; BCL, B Cell Lymphoma; CPCs, Cardiac Progenitor cells; DOX, Doxorubicin; E2F1, E2F Transcription factor 1;Foxo, forkhead box O; GSH, Glutathione; HO, Heme Oxygenase; LC3, Light chain 3; LV, left ventricular; MAPK, Mitogen-activated protein kinase; MDA, Malondialdehyde; Mn-SOD, Manganese superoxide dismutase; mTORC1, Mammalian target of Rapamycin Complex 1; MuRF-1, Muscle RING-finger protein-1; NO, Nitric oxide; NFAT3, Nuclear Factor of Activated T-Cells; PI3K, phosphatidylinositol 3-kinase; RESV, Resveratrol; SIRT1, Sirtuin 1; SOD, Superoxide Dismutase; TGF-β, transforming growth factor-β; ↑, increase; and ↓, decrease.

**Table 3 nutrients-11-00627-t003:** In vitro studies demonstrating anticancer effect of the combination of resveratrol and anthracycline treatment.

Study	Cancer Cell Type	DOX Treatment (Conc./Time)	Resveratrol Treatment (Conc./Time/Time Relative to DOX)	Finding	Proposed Mechanism
Fulda S et al. 2004 [107]	U373MG glioblastoma MCF-7 breast cancer cells LNCaP prostate carcinoma Reh B-cell leukemia cells	0–0.1 μg/mL (0–0.18 μM) for 24 h	30 μM	RESV significantly enhanced DOX-induced apoptosis in a dose-dependent manner	Inducing cell cycle arrest which led to survivin depletion
Rezk YA et al. 2006 [59]	Human ovarian cancer cells OVCAR-3 and uterine (Ishikawa) cells	10 nM to 100 μM	1 μM 1 h pre-DOX for 48 h	Combination with RESV demonstrated an additive growth-inhibitory effect	RESV pro-apoptotic effects
Gatouillat G et al. 2010 [108]	Mouse chemo-resistant B16 melanoma	0.25–5 μM for 24 h	10 or 25 μM pretreatment for 24 h with an additional 24 h	Pretreatment with RESV resulted in increased cytotoxicity of DOX	↑ p53 expression ↑ Apoptosis Induce cell cycle arrest in S-Phase
Al-Abd AM et al. 2011 [109]	Human Hepatocellular carcinoma cell line (HepG2) Breast cancer cell line (MCF-7) Cervical cancer cell line (HeLa)	0.001–100 μg/mL (1.8 Nm–184 μM)	0.001–100 μg/mL (4.4 nM–440 μM)	Synergistic interaction between RESV and DOX on MCF-7 cells, and additive interactions on HeLa and HepG2 cells	↓ P-gp and MDR1 expression ↑ Bax expression
Osman AM et al. 2012 [110]	Human breast cancer cell line MCF-7	0.0312–5 μg/mL (57 nM–9 μM) for 48 h	15 μg/mL (65 μM) either simultaneously for 48 h or started 24 h pre-DOX	Both approaches increased the cytotoxic activity of DOX	Simultaneous treatment ↑ DOX cellular accumulation Pretreatment ↓ DOX cellular accumulation
Diaz-Chavez J et al. 2013 [111]	DOX resistant MCF-7, MDA-MB-231	5 μM for 48 h	100, 150, 200, and 250 μM	RESV induced apoptosis and sensitization to DOX	Inhibition of HSP27 expression Disruption of mitochondrial membrane potential Cytochrome C release
Kim TH et al. 2014 [112]	DOX-resistant breast cancer (MCF-7/adr) and MDA-MB-231 cells	1 μM for 12 h	50 μM for 12 h	RESV increased the efficacy of Dox in human breast cancer cells	↑ DOX cellular accumulation and decrease resistance ↑ ATP-binding cassette (ABC) transporter genes, MDR1, MRP1
Schroeter A et al. 2014 [113]	HT-29 human colon carcinoma cells	10 μM	50–250 μM pretreatment for 30 min then 1 h co-administration	RESV affected the TOP-poisoning potential of DOX and modulated its cytotoxic effectiveness	RESV counteracted DOX-induced formation of DNA-TOP-intermediates at ≥100 μM for TOP IIα and at 250 μM for TOP IIβ ↓ Mitochondrial activity >200 μM RESV ↓ the intracellular DOX concentration
Huang F et al. 2014 [114]	Human breast cancer DOX-resistant (MCF-7/ADR) cells	0.01, 0.1, 1, 10, or 100 μM for 48 h	4, 8, 12, or 16 μM for 48 h	RESV sensitized MCF-7/ADR cells to DOX	↑ DOX intracellular concentration, ↓ P-gp and MDR1 expression
Tomoaia G et al. 2015 [115]	HeLa and Caski Cells	1.5 μM	200 μM	RESV increased sensitivity to DOX and decreased cell viability	↑ Apoptosis
Sheu MT et al. 2015 [116]	MCF-7 breast cancer cells	1.5 μM for 48 h	200 μM RESV in combination with curcumin, THSG, and EEAC for 48 h	RESV demonstrated synergistic effects with Dox	Antioxidant properties of RESV
Khaleel, SA et al. 2016 [117]	HCT 116 and HT-29	Serial concentration of DOX for 72 h	Serial concentration of RESV for 24 h	RESV sensitized colorectal cancer cells to DOX	↑ Bax gene expression ↑ p53 gene expression Induction of S-phase arrest ↑ Intracellular DOX entrapment by blocking P-gp
Hashemzaei M et al. 2016 [118]	Lymphoblastic leukemia cell line (MOLT-4) Human multiple myeloma cell line (U266B1) Burkitt’s Lymphoma cell line (Raji cell)	Daunorubicin 0.5, 0.5, and 0.7 μM for 3 h for each cell line, respectively	20, 73, and 47 μM for 72 h for each cell line, respectively	Combination of RESV and DAN significantly increased cell death in the three cell lines	↑ Apoptosis
Rai G et al. 2016 [119]	Human breast cancer cell lines (MDA-MB-231 and MCF-7)	312 and 781 nM in MDA-MB-231 216 and 541 nm in MCF-7	108 and 180 μM in MDA-MB-231 84.6 and 141 μM in MCF-7	RESV enhanced the cytotoxic effect of DOX	↑ Apoptosis Suppression of chronic inflammation markers (NF-κB, COX-2) and autophagy (LC3B, Beclin-1)
Xu J et al. 2017 [120]	DOX-resistant gastric cancer cells (SGC7901/DOX)	1 mg/L (1.8 μM) for 48 h	50 mg/L (220 μM) for 48 h	RESV restored DOX sensitivity and promoted cell apoptosis ↓ Cell migration	Reversal of EMT ↓ Vimentin and β-catenin ↑ E-cadherin Modulating PTEN/Akt signaling pathway ↑ PTEN and caspase-3
Chen JM et al. 2018 [121]	DOX-resistant human breast carcinoma cells (MCF-7/DOX)	1 mg/L (1.8 μM) for 7 days	10 mg/L (43 μM) for 7 days	RESV increased DOX-induced cell apoptosis, inhibited cell proliferation, decreased cell migration and reversed DOX resistance	Modulating PI3K/Akt signaling pathway ↑ PI3K and caspase-3 ↓ p70 S6K and p-Akt/Akt ratio
Barros AS et al. 2018 [122]	PANC-1 cells in 2D and 3D culture models	Three different concentrations of DOX: RESV (20.1, 46.3, and 106.3 μM) with different molar ratios (ranging from 5:1 to 1:5) for 24 h		DOX: RESV combination induced the highest reduction on PANC-1 cells viability	↓ P-gp ↑ Intracellular accumulation of DOX
Carlson A et al. 2018 [123]	Canine Hemangiosarcoma Cells	1 μM for 22 or 46 h	10-100 μM for 24 or 48 h	RESV increased the growth inhibitory effect of DOX in hemangiosarcoma cells	↑ DOX-induced apoptosis
El-Readi MZ et al. 2018 [124]	Resistant cell lines: Caco-2 and CEM/ADR5000	0.1–500 μM	20 μM	RESV significantly increased DOX cytotoxicity	Inhibition of ABC-transporters (P-gp, MDR1, and MRP1)
Pouyafar A et al. 2019 [125]	Cancer stem cells (CSCs) isolated from ovarian carcinoma cell line (SKOV3)	250 nM for 24 and 48 h	55 μM for 24 and 48 h	RESV with DOX diminished the cell viability and decreased drug resistance to DOX	↑ DOX-induced apoptosis ↑ BAX expression ↓ MDR1 and MRP1 genes expression
Jin X et al. 2019 [126]	DOX resistant human breast cancer MCF7/ADR	4 μg/mL (7.4 μM) for 48 h	50 μM	RESV increased sensitivity to DOX and inhibited cell growth	Induce cell apoptosis ↓ Cell migration ↓ DOX-induced upregulation of Vimentin, N-cadherin, and β-catenin ↑ Sirt1

ABC, ATP-binding cassette; ADR, Adriamycin; Akt, Protein Kinase B; Bax, BCL (B Cell Lymphoma)-Associated X; BCL, B Cell Lymphoma; Cox-2, cyclooxygenase-2; DAN, Daunorubicin; DOX, Doxorubicin; EMT, epithelial-mesenchymal transition; HSP27, Heat shock protein 27; LC3, Light chain 3; MDR1, Multidrug Resistance-1; MRP1, Multidrug resistance-associated protein-1; MuRF-1, Muscle RING-finger protein-1; NF-κB, nuclear factor kappa B; NO, Nitric oxide; PI3K, phosphatidylinositol 3-kinase; P-gp, P-glycoprotein; PTEN, Phosphatase and tensin homolog; RESV, Resveratrol; TOP, topoisomerase; ↑, increase; and ↓, decrease.

**Table 4 nutrients-11-00627-t004:** In vivo studies demonstrating anticancer effect of resveratrol co-administration with anthracyclines.

Study	Species/Strain	Cancer Type	DOX Treatment (Dose/Time)	Resveratrol Treatment (Dose/Time/Time Relative to DOX)	Finding	Proposed Mechanism
Kim TH et al. 2014 [112]	Female BALB/c athymic nude mice	Dox-resistant breast cancer (MCF-7/ADR)	4 mg/kg i.p. once weekly for 4 weeks	20 mg/kg with oral gavage (once weekly) for 4 weeks	RESV and Dox combination significantly reduced tumor volume by approximately 50% compared to the control (No RESV only group)	RESV significantly diminished the levels of MDR1 or MRP1
Rai G et al. 2016 [119]	Male Swiss albino mice	Ehrlich Ascitic Carcinoma (EAC) cells	5 mg/kg i.p. twice weekly	10 mg/kg i.p.	RESV and DOX could be used as a novel combination therapy for human breast cancer	Induced apoptosis ↑ Intracellular ROS
Xu J et al. 2017 [120]	Nude mice	SGC7901/DOX xenografts	3 mg/kg i.p. once weekly for 4 weeks	50 mg/kg once weekly by oral gavage	RESV and dox synergistically inhibited tumor growth	↑ PTEN and cleaved-caspase-3 ↓ Vimentin and ki67
Chen JM et al. 2017 [121]	Nude mice	MCF-7/DOX cells	3 mg/kg i.p. once weekly for 4 weeks	50 mg/kg with oral gavage	RESV with DOX synergistically reduced the tumor growth	↓ P-Akt, p70 S6K and ki67 expression
Hallajian F et al. 2018 [127]	Nu/b6 nude mice	MCF-7, human breast adenocarcinoma	6 doses 2.5 mg/kg i.p. (cumulative dose of 15 mg/kg over 2 weeks)	20 mg/kg/day for 2 weeks	Combination of DOX and RESV had shown a higher antitumor effect. Moreover, it reduced the cardiotoxicity and hepatotoxicity induced by DOX	↑ DOX-Induced tumor apoptosis and necrosis

Akt, Protein Kinase B; DOX, Doxorubicin; MDR1, Multidrug Resistance-1; MRP1, Multidrug resistance-associated protein-1; PI3K, phosphatidylinositol 3-kinase; PTEN, Phosphatase and tensin homolog; RESV, Resveratrol; ROS, Reactive Oxygen Species; ↑, increase; and ↓, decrease.

**Table 5 nutrients-11-00627-t005:** Studies demonstrating the radiomodulatory effects of resveratrol.

Study	Cancer Type/Cell Line	Radiation Dose	Resveratrol Dose/Conc.	Major Findings	Proposed Mechanisms
Zoberi I et al. 2002 [216]	Human cervical carcinoma cell lines, HeLa and SiHa	Ionizing Radiation (IR) 2–8 Gy	10 or 25 μM for 4–48 h	RESV enhanced tumor cell killing by IR in a dose-dependent manner	Inhibition of COX-1
Baatout S et al. 2004 [217]	HeLa (cervix carcinoma), K-562 (chronic myeloid leukemia) IM-9 (multiple myeloma)	0–8 Gy	0–200 μM started 1 h before the irradiation	RESV can act as a radiation sensitizer at high concentrations	Induced apoptosis and inhibition of cell growth
Baatout S et al. 2005 [218]	Human leukemic cell line, EOL-1	0, 2, 4, 6, or 8 Gy	0–200 μM	Depending on the concentration, RESV could enhance radiation-induced apoptosis	Induced apoptosis and inhibition of cell growth
Liao HF et al. 2005 [219]	Human non-small cell lung cancer NCI-H838	0.5, 1, 2, and 3 Gy	6.25–50 μM	RESV sensitized NCI-H838 to radiation in a concentration-dependent manner	NF-κB inhibition and S-phase arrest
Scarlatti F et al. 2007 [220]	DU145 prostate cancer cells	0.5–2.0 Gy/day for 3 consecutive days	0.5–32 μM for 72 h	RESV significantly enhanced radiation-induced cell death	RESV potentiated ionizing radiation-induced ceramide accumulation, by promoting its de novo biosynthesis
Lu KH et al. 2009 [221]	Medulloblastoma-associated cancer stem cells	Ionizing radiation 0–10 Gy	0, 10, 50, 100, and 150 μM for 48 h	RESV 100 μM enhanced the radiosensitivity	Anti-proliferative properties
Kao CL et al. 2009 [222]	CD133-positive/negative cells derived from atypical teratoid/rhabdoid tumors	2 Gy	150 μM	RESV enhanced IR-mediated apoptosis	↓ Proliferation
Rashid A et al. 2011 [223]	Androgen-insensitive (PC3), sensitive (22RV1) prostate cancer cells	IR (2–8 Gy)	2.5–10 μM	RESV enhanced IR-induced nuclear aberrations and apoptosis in cancer cells	RESV enhanced IR activation of ATM and AMPK but inhibited basal and IR-induced phosphorylation of Akt
Fang Y et al. 2012 [224]	Prostate cancer cell line, PC-3	2–8	2–10 μM	RESV augmented radiation-induced inhibition of cell proliferation and reduction of cell survival	Increased apoptosis and senescence
Yang YP et al. 2012 [225]	Primary Glioblastoma cells	In vitro: 0, 2, 4, 6, 8, and 10 Gy	100 μM for 24 h	RESV induced apoptosis and enhanced radiosensitivity of glioblastoma cells	Inhibiting the STAT3 Axis
Tak JK et al. 2012 [226]	Mouse colon carcinoma CT26 and mouse melanoma B16F10 cells	15 Gy γ-irradiation	10 and 20 µM	RESV sensitized the cancer cells to radiation-induced apoptosis	Increased ROS
Fang Y et al. 2012 [227]	Prostate cancer cell lines (PC-3 cells and DU145 cells)	2, 4, and 8 Gy	0–50 μM for 24 h	Combination of radiation and RESV additively/synergistically decreased survival of PCA	↑ Apoptosis ↑ Perforin and granzyme B expression
Yang YP et al. 2012 [225]	Glioblastoma multiforme (GBM)-derived CD133+ radioresistant tumor-initiating cells (TIC)	0, 2, 4, 6, 8, and 10 Gy	100 μM for 24 h	RESV could significantly improve the survival rate and synergistically enhance radiosensitivity of radiation-treated GBM-TIC	STAT3 Pathway by suppressing STAT3 signaling. ↓ Bcl-2 and survivin expression
Fang Y et al. 2013 [228]	Radio-resistant human melanoma lines (SK-Mel-5 and HTB-65)	1, 2, and 4 Gy	0–50 μM for 24 h	RESV enhanced radiation sensitivity of melanoma cells	↓ Proliferation ↓ Expression of proproliferative molecules (cyclin B, cyclin D, cdk2 and cdk4) ↑ Apoptosis ↓ Expression of anti-apoptotic molecules (FLIP, Bcl-2, and survivin)
Luo H et al. 2013 [229]	Human non-small cell lung cancer (NSCLC) cell lines A549 and H460	(0–8 Gy) of irradiation at 4 h after RESV pre-treatment	20 μM	RESV sensitized the cancer cells to radiation-induced cell death	Enhancing IR-induced premature senescence via increasing ROS-mediated DNA damage
Magalhães VD et al. 2014 [230]	Human rhabdomyosarcoma cells (RD)	50 and 100 Gy	15, 30, and 60 μM for 24 h	RESV 15 μM protected cells and had cytotoxic effect at 60 μM	RESV at 60 μM could inhibit cell growth and induce apoptosis
Heiduschka G et al. 2014 [231]	Merkel cell carcinoma MCC13 and MCC26 cells	1, 2, 3, 4, 6, and 8 Gy	7.5 and 15 μM	RESV and irradiation led to synergistic reduction in colony formation compared to irradiation alone	No specified mechanism was mentioned
Antienzar AN et al. 2014 [232]	Oral squamous cell carcinoma PE/CA-PJ15 cells	1, 2.5, and 5 Gy	5, 10, 25, 50, and 100 μM for 24, 48, and 78 h	RESV increased radiation-induced cell death, apoptosis and migration in conc and time-dependent manner	Induce apoptosis and cell migration
Wang L et al. 2015 [233]	Glioma stem cell line, SU-2 For the in vivo studies: Five-week-old male nude (BALB/c) mice	In vitro: 0, 2, 4, 6 Gy In vivo: IR (X-ray, 6 Gy) twice on day 3 and day 9	In vitro: 75 μM In vivo: 150 mg/kg every other day for 2 weeks	RESV enhanced radiation-induced effects both in vitro and in vivo	Inhibition of self-renewal and stemness Induction of autophagy, promotion of apoptosis, and prevention of DNA repair
Baek SH et al. 2016 [234]	Squamous cell carcinoma of the head and neck FaDu cells	1, 5, and 10 Gy	50 and 100 2 h pretreatment and 24 h after radiation	RESV potentiated the effect of radiation on FaDu cells	Inhibition of STAT3 signaling pathway through the induction of SOCS-1
Chen YA et al. 2017 [235]	Prostatic cancer LAPC4-KD cells In vivo: Male nude BALB/c mice	In vitro: 2 Gy In vivo: 12 Gy delivered in 3 doses on day 0, 3, and 7	In vitro: 25 μg/mL (100 μM) for 24 h In vivo: 5 mg/kg	RESV inhibited the proliferation and increased radiosensitivity of radioresistant cells. Moreover, RESV inhibited tumor growth in vivo	Induce apoptosis ↑ Caspase-3
Tan Y et al. 2017 [236]	Nasopharyngeal cancer, CNE-1 cells in vitro BALB/c nude mice in vivo	In vitro: 0, 2, 4, 6 Gy In vivo: 4 Gy	In vitro: 50 μM for 24 h In vivo: 50 mg/kg/day On day 8, mice were treated i.p. with RESV or vehicle until the completion of the experiment. On day 12, mice were irradiated once a day for consecutive 3 days. Mice were sacrificed on day 28 to measure tumor volume and tumor weight	RESV sensitized CNE-1 cells to radiation in vitro and in vivo	Downregulating E2F1 and inhibiting p-AKT
Ji K et al. 2018 [215]	Human lung cancer cell lines A549 and H460	0, 2, 4, or 6 Gy irradiation	50 µM for 24 h	RESV reduced radiation-induced apoptosis (protected the cancer cells from radiation)	Activation of Sirt1 protects cancer cells from radiation
Voellger B et al. 2018 [212]	Rodent GH3 and TtT/GF pituitary adenoma cells	0–5 Gray	10 and 100 μM for 2 h before radiation and further incubated for 48–72 h	Combination of RESV and irradiation significantly decreased cell viability	Induce cell death
Banegas YC et al. 2018 [211]	Human lung cancer cell line A549	4 and 16 Gy X-ray	15 and 60 μM for 24 and 120 h	RESV 60 μM had a radiosensitizing effect	RESV 15 μM had no effect on tumor cells, but was radioprotective in CHO-k1 cells

Akt, Protein Kinase B; AMPK, Adenosine Monophosphate-Activated Protein Kinase; ATM, Ataxia-Telangiectasia Mutated; BCL, B Cell Lymphoma; Cdk, cyclin-dependent kinase; E2F1, E2F Transcription factor 1; FLIP, FLICE-like inhibitory protein; IR, ionizing radiation; NF-κB, nuclear factor kappa B; PCA, prostate cancer; Sirt1, Sirtuin 1; SOCS, Suppressor of cytokine signaling; STAT3,Signal transducer and activator of transcription 3; IR, Ionizing Radiation; RESV, Resveratrol; ROS, Reactive Oxygen Species; ↑, increase; and ↓, decrease.

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
