# Peer review of "Leveraging the Cardio-Protective and Anticancer Properties of Resveratrol in Cardio-Oncology"

_nutrients, 2019, doi:10.3390/nu11030627_

Reviewer 1 Report

The manuscript “Leveraging the Cardio-Protective and Anti-Cancer Properties of Resveratrol in Cardio-Oncology” submitted by Ibrahim Y. Abdelgawad et al. provides a comprehensive and critical revision of the use of resveratrol in the protection of the cardiovascular system during cancer treatment. In the manuscript, the author reviewed the current state of the art of the different preclinical studies carried out up to date and discuss the challenges that face the clinical application of this agent. Overall, the manuscript is well presented, organized and the topic reviewed is interesting for the field. However, I recommend the incorporation of additional information to improve the final quality of the work.

Minor comments:

1)     I recommend the incorporation of a general section in the manuscript introducing resveratrol and summarizing the pharmacokinetic properties of this compound, as well as the nutritional sources, cytotoxicity, routes of administration studied, bioavailability, and the molecular effects reported for this drug in cellular and animal models. This might help to improve the understanding of the role of this molecule in cardio-protection and cancer treatment in the subsequent sections.

2)     Related to the point above, the incorporation of a figure showing the chemical structure of resveratrol (and its chemical analogs) is also highly recommended. As a summary, this figure can also include a panel describing the main roles of resveratrol in cardio-oncology.  

Author Response

Response to Reviewer 1 Comments:

The manuscript “Leveraging the Cardio-Protective and Anti-Cancer Properties of Resveratrol in Cardio-Oncology” submitted by Ibrahim Y. Abdelgawad et al. provides a comprehensive and critical revision of the use of resveratrol in the protection of the cardiovascular system during cancer treatment. In the manuscript, the author reviewed the current state of the art of the different preclinical studies carried out up to date and discuss the challenges that face the clinical application of this agent. Overall, the manuscript is well presented, organized and the topic reviewed is interesting for the field. However, I recommend the incorporation of additional information to improve the final quality of the work.

Response: We would like to thank the reviewer for her/his favorable appraisal of our submitted review and for the constructive critique that helped us improve the quality of the work. Herein, we provide a point by point response to the raised concerns.

Minor comments:

1)     I recommend the incorporation of a general section in the manuscript introducing resveratrol and summarizing the pharmacokinetic properties of this compound, as well as the nutritional sources, cytotoxicity, routes of administration studied, bioavailability, and the molecular effects reported for this drug in cellular and animal models. This might help to improve the understanding of the role of this molecule in cardio-protection and cancer treatment in the subsequent sections.

Response: We largely agree with the reviewer about the importance of such introduction. In the revised submission, we discussed important pharmacological properties of resveratrol early on in the introduction section, so that the rest of the review would be better understood. We have also referred to excellent reviews that comprehensively discussed the pharmacology of resveratrol.

Please, refer to lines 47-78:

“Resveratrol (3,5,4'-trihydroxy-trans-stilbene) is a polyphenolic compound that is naturally occurring in several plant based foods and beverages including red wine, grapes, berries, and peanuts [11]. In addition, one of the richest sources of resveratrol is Polygonum cuspidatum, an herb that plays an important role in Chinese traditional medicine [12].  Indeed, resveratrol is a phytoalexin produced by more than 70 species of plants in response to stressful situations such as infection [13]. Resveratrol has been a focus of scientific research over the past 25 years, primarily as an active ingredient in red wine that is responsible for the “French Paradox” [14]. The “French Paradox” describes the low incidence of cardiovascular diseases in the French population despite high consumption of saturated fats [15]. Nevertheless, the concentration of resveratrol in red wine and other plants is very low and research into resveratrol has mostly used high doses that can only be achieved by supplementary intake of resveratrol as a nutraceutical [16]. Additionally, orally administered resveratrol shows very low bioavailability despite almost complete absorption (reviewed in [17]). Resveratrol low bioavailability is due to its first-pass intestinal/hepatic metabolism to sulfate and glucuronide metabolites [18]. After an oral dose of 25 mg in healthy human volunteers, the plasma concentration of resveratrol was less than 10 µg/L (≈40 nM), while the total concentration of resveratrol and its metabolites was 400-500 µg/L (≈2 µM) [18,19]. Higher peak plasma concentrations of resveratrol (approximately 500-2000 µg/L ≈2-10 µM) were achieved with doses of 2000-5000 mg [20-22]. Accordingly, the findings of in vitro studies using resveratrol concentrations higher than 10 µM should be interpreted with caution.

Preclinical in vitro and in vivo studies have revealed that resveratrol possesses a plethora of beneficial effects in an array of diseases (comprehensively reviewed in [23]), including: cardiovascular diseases (reviewed in [24-26]), cancer (reviewed in [27,28]), obesity (reviewed in [29]), osteoporosis (reviewed in [30]), Alzheimer’s disease (reviewed in [31]), and diabetes (reviewed in [32]). Resveratrol has also gained considerable attention as an anti-aging molecule (reviewed in [33]). It may be surprising that a single molecule can have beneficial effects in such a wide variety of diseases. Indeed, this may be attributed to the multiple molecular targets of resveratrol (reviewed in [34]). Resveratrol has been shown to activate AMP-activated protein kinase (AMPK) [35], sirtuin 1 (SIRT1) [36,37], super oxide dismutase (SOD) [38], nuclear factor erythroid 2–related factor 2 (NRF2) [39], vascular endothelial growth factor (VEGF) [40], and endothelial nitric oxide synthase (eNOS) [37,41]. It has also been shown to inhibit cyclooxygenases [42], phosphodiesterases  (PDEs) [43], nuclear factor kappa B (NF-κB) [44,45], aryl hydrocarbon receptor (AhR) [46], phosphoinositide 3-kinase (PI3K) [47], mammalian target of rapamycin (mTOR) [48], and ribosomal protein S6K [49]”.

2)     Related to the point above, the incorporation of a figure showing the chemical structure of resveratrol (and its chemical analogs) is also highly recommended. As a summary, this figure can also include a panel describing the main roles of resveratrol in cardio-oncology.  

Response: We greatly appreciate this excellent suggestion. In the revised submission, we have added a figure with the chemical structure of resveratrol, a summary of its cardio-protective and anti-cancer properties, and the underlying mechanisms. We believe that this figure is a valuable addition to the review.

Please, refer to Figure 1 in the revised submission.

Reviewer 2 Report

With the new therapeutic treatments for cancer patients increased the survival time, the adverse cardiovascular events caused by the treatments become clinically evident. The study of cardio-protective strategies derived from the observations that a group of compounds known as anthracyclines caused heart failure. In this review, the authors discussed the cardio-protective and anti-cancer properties of resveratrol. They also discussed the challenges of the application of resveratrol and its analogs. The review well summarized the current studies of resveratrol’s protection against cardiotoxicity. The comments on this review are as followed:

Currently, the tools, such as strain echocardiography, cardiac MRI and the increased use of biomarkers, used to measure cardiac function are not very effective. The development of new methods to recognize the cardio damage early during or after cancer treatments might help oncologists decide whether to use cardio-protective agents. The authors may add some information on the diagnosis of cardiovascular diseases to Point 5 “Challenges to the Clinical Use of Resveratrol in Cardio-Oncology”.

Some studies showed that resveratrol modulates human immune cell functions. This might be one of the reasons that resveratrol augments anticancer treatments.

Author Response

Response to Reviewer 2 Comments:

With the new therapeutic treatments for cancer patients increased the survival time, the adverse cardiovascular events caused by the treatments become clinically evident. The study of cardio-protective strategies derived from the observations that a group of compounds known as anthracyclines caused heart failure. In this review, the authors discussed the cardio-protective and anti-cancer properties of resveratrol. They also discussed the challenges of the application of resveratrol and its analogs. The review well summarized the current studies of resveratrol’s protection against cardiotoxicity. The comments on this review are as followed:

Response: We would like to thank the reviewer for his/her favorable appraisal of our submitted review and for the constructive critique that helped us improve the quality of the work. Herein, we will provide a point by point response to the raised concerns.

Currently, the tools, such as strain echocardiography, cardiac MRI and the increased use of biomarkers, used to measure cardiac function are not very effective. The development of new methods to recognize the cardio damage early during or after cancer treatments might help oncologists decide whether to use cardio-protective agents. The authors may add some information on the diagnosis of cardiovascular diseases to Point 5 “Challenges to the Clinical Use of Resveratrol in Cardio-Oncology”.

Response: We totally agree with the reviewer about this point that is a general challenge in the field of cardio-oncology. We have added this point to the “Challenges” section as suggested by the respected reviewer. Please, refer to lines 404-408.

“Furthermore, the predictive and diagnostic value of different approaches such as strain echocardiography, cardiac magnetic resonance imaging, and biomarkers are still not well established in the field of cardio-oncology [282]. Therefore, it is difficult to identify those patients who will benefit the most from a potential cardio-protective therapy”.

Some studies showed that resveratrol modulates human immune cell functions. This might be one of the reasons that resveratrol augments anticancer treatments.

Response: This is also an important point. In fact, most of the in vivo cancer models discussed in the review employed immunocompromised mouse models which may have obscured this important effect of resveratrol. We have discussed this limitation in the revised submission. Please, refer to lines 192-196.

“A major limitation to the vast majority of these studies is the use of immunocompromised nude mice implanted with human cancer cells (Table 4). This approach precludes studying the immunomodulatory properties of resveratrol which are increasingly recognized to be important players in its chemo-preventive and anti-cancer effects [128-130]”.